# 5-HT6 Receptors Sex-Dependently Modulate Hippocampal Synaptic Activity through GABA Inhibition

**DOI:** 10.3390/biom13050751

**Published:** 2023-04-26

**Authors:** Caroline Lahogue, Jean-Marie Billard, Thomas Freret, Valentine Bouet

**Affiliations:** Department of Health, UNICAEN, INSERM, COMETE, CYCERON, FHU A^2^M^2^P, CHU Caen, Normandie Université, 14000 Caen, France

**Keywords:** hippocampus, serotonin, electrophysiology, NMDA, long-term potentiation

## Abstract

The subtype 6 of the serotoninergic receptors (5-HT6Rs) is highly expressed in the hippocampus, and evidence indicates the beneficial effects of 5-HT6Rs blockade on short- and long-term memory in rodents. Nevertheless, the underlying functional mechanisms still need to be established. To this end, we performed electrophysiological extracellular recordings to assess the effects of the 5-HT6Rs antagonist SB-271046 on the synaptic activity and functional plasticity at the CA3/CA1 hippocampal connections of male and female mice slices. We found that basal excitatory synaptic transmission and isolated N-methyl-D-aspartate receptors (NMDARs) activation were significantly increased by SB-271046. The NMDARs-related improvement was prevented by the GABAAR antagonist bicuculline in male but not in female mice. Regarding synaptic plasticity, neither paired-pulse facilitation (PPF) nor NMDARs-dependent long-term potentiation (LTP) (induced either by high-frequency or theta-burst stimulation) was affected by the 5-HT6Rs blockade. Taken together, our results indicate a sex-dependent 5-HT6Rs effect on synaptic activity at the CA3/CA1 hippocampal connections through changes in the excitation/inhibition balance.

## 1. Introduction

The serotoninergic subtype 6 receptors (5-HT6Rs) are mainly expressed in the central nervous system, particularly in brain regions involved in memory processes [1,2,3,4,5]. This distribution can support a role for these receptors in the treatment of memory disorders related to neuropsychiatric diseases such as those observed in schizophrenia. Although many typical and atypical antipsychotics have an affinity for 5-HT6Rs [6,7,8,9] and alleviate positive symptoms (hallucinations and delusions), negative symptoms (avolition, anhedonia, etc.) and cognitive deficits (working memory, attentional deficit, etc.) of schizophrenia remain resistant, and several side effects are currently reported in schizophrenic patients [10]. Therefore, due to their localization in the central nervous system, suggesting the absence of adverse peripheral effects, 5-HT6Rs are interesting targets [11]. Consequently, the development of pharmacological agents targeting the 5-HT6Rs is currently ongoing in both clinical and preclinical studies [12,13,14]. Several studies have reported beneficial effects of 5-HT6Rs blockade on behavior. For instance, studies in rats and mice have shown beneficial effects of 5-HT6Rs’ blockade to reverse scopolamine-induced memory deficits [15,16,17,18] and to reverse memory impairment related to aging in mice [19]. In addition, several studies have shown beneficial effects of 5-HT6Rs’ blockade on the memory and learning performance in animal models of schizophrenia [20,21]. Taken together, the preclinical results of 5-HT6Rs blockade show beneficial effects on cognition processes. Interestingly, improved performance has been reported in hippocampal-dependent memory tasks after the blockade of 5-HT6Rs [15,16,19,21,22]. From a paradoxical point of view, while 5-HT6Rs’ activation by agonists leads to impaired flexibility, working memory, and social recognition in rodents [17,23], other studies have shown that their activation could improve recognition memory and decreased anxiety-related behavior [18,24]. Furthermore, cognitive impairments (spatial learning in Morris water maze) and increased anxiety level were found in 5-HT6Rs null mutant mice [25].

Despite the pro-cognitive evidence of 5-HT6Rs antagonism, the cellular mechanisms underlying the memory improvement are still misunderstood. Several behavioral studies have highlighted the involvement of serotonin–acetylcholine interactions [15,16,17,21,26] or serotonin-glutamate neurotransmission systems interactions [21,27] in memory processes. Microdialysis studies showed that the blockade of 5-HT6Rs by SB-271046 increases glutamate and acetylcholine release in the hippocampus [28,29,30,31]. Conversely, 5-HT6Rs activation with the agonists WAY-181187 or WAY-208466 increases extracellular level of γ-Aminobutyric acid type A (GABA) without affecting those of serotonin, dopamine and glutamate in the dorsal hippocampus [32]. These neurochemical data are in agreement with the localization of 5-HT6Rs on GABAergic interneurons in several brain areas such as the hippocampus [3,33,34,35,36]. Based on these data, it has been suggested that 5-HT6Rs could directly increase GABAergic activity and indirectly increase glutamatergic neurotransmission [3,32]. 

Functional synaptic plasticity within hippocampal networks is a critical mechanism for memory processes. In particular, long-term potentiation (LTP) dependence on synaptic transmission due to activation of the N-methyl-D-aspartate subtype of glutamate receptors (NMDARs) plays a key role [37,38,39,40,41]. Yet, the effect of 5-HT6Rs’ modulation of hippocampal NMDARs-dependent LTP remains poorly considered [42]. This study therefore aimed to functionally characterize the interplay between 5-HT6Rs and glutamatergic neurotransmission by assessing the effects of the 5-HT6Rs’ antagonist SB-271046 on synaptic activity and plasticity of CA3/CA1 hippocampal synapses. SB-271046, whose structure and function have been initially described by Bromidge et al. (1999) [4] and Routledge et al. (2000) [5], is known to block the 5-HT6Rs, leading to a decrease in adenylyl-cyclase activation and a decrease in AMPc intracellular concentration. Our investigation was performed in both male and female mice, allowing us to take into consideration the sexual dimorphism that can be observed in healthy brains [43] but is also accentuated in diseases such as schizophrenia [44,45,46]. In addition to morphological and symptomatology dimorphisms [44,45,46], the effective dosage of antipsychotic drug to obtain a response is different between male and female patients, with male patients requiring significantly higher amounts [47,48]. Interestingly, a clinical investigation using the 5HT6Rs’ agonist inhibitor avisetron (AVN-211) found a sex-dependent response in schizophrenia patients with residual psychotic symptoms [49], which reinforces the importance of testing the effect of the molecule on male and female mice.

## 2. Materials and Methods

### 2.1. Animals

In the present study, 16 male and 13 female C57BL/6 mice, aged 5–8 months, provided by our local facility (Centre Universitaire de Ressources Biologiques, Université de Caen, Normandie, France), were used. Mice were kept in polycarbonate cages (n = 5 per cage) in standard controlled conditions including reversed 12/12 h light/dark cycle (lights on at 7:30 a.m.) and stable temperature (22 ± 2 °C) and humidity (50 ± 10%). Mice were given free access to food and water. Cage enrichment consisted of crinkle-cut shredded paper and a carboard house. Experiments were performed in accordance with French and European Economic Community guidelines for the care and use aof laboratory animals (Directive 2010/63/UE, project no. 2020121013375130 v2).

### 2.2. Pharmacology

Based on its pharmacological profile [4], the selective 5-HT6Rs’ antagonist SB-271046 (5-chloro-N-(4-methoxy-3-piperazin-1-yl-phenyl)-3-methyl-2-benzothiophene sulfonamide hydrochloride) was first delivered at concentrations of 0.1 µM, 1 µM, or 10 µM to assess the dose-dependent effect on NMDARs’ activation. The higher dose was then selected to investigate the effects of SB-271046 on non-NMDARs’ activation and the expression of synaptic plasticity. Both NBQX (2,3-Dioxo-6-nitro-1,2,3,4-tetrahydrobenzo [f]quinoxaline-7- sulfonamide, NBQX) and bicuculline (non-NMDARs and γ-aminobutyric acid type A receptor (GABAAR) antagonists, respectively) were used at 10 µM. SB-271046 was synthesized by F. Fabis and C. Fossey (Centre d’Études et de Recherche sur le Medicament de Normandie, CERMN, UNICAEN), NBQX was purchased from Tocris Bioscience^®^ (Bristol, UK) and bicuculline from Sigma-Aldrich^®^ (Lyon, France).

### 2.3. Extracellular Electrophysiology

Mice were anesthetized with 5% isoflurane and decapitated. The brain was rapidly extracted from the skull and placed in ice-cold artificial cerebrospinal fluid (aCSF) containing 124 mM NaCl, 3.5 mM KCl, 1.5 mM MgSO_4_, 26.2 mM NaHCO_3_, 1.2 mM NaH_2_PO_4_, 2.5 mM CaCl_2_, and 11 mM glucose (pH 7.4). The solution was gassed with an O_2_/CO_2_ mixture (95%/5%). Transverse hippocampi slices (400 µm thick) were obtained with a tissue slicer (Mc ILwain^®^, Redding, CA, USA) and placed in aCSF for at least 60 min in a holding chamber at room temperature before recording. The hippocampal slice was then placed into the recording chamber, maintained with a grid, and continuously submerged at room temperature with pregassed aCSF.

Specific NMDAR activation was assessed in the CA1 hippocampal area by conventional single-array recordings using glass micropipettes (2–5 MΩ) filled with 2 M NaCl. Presynaptic fiber volleys (PFVs) and field excitatory postsynaptic potentials (fEPSPs) were evoked at 0.1 Hz by the electrical stimulation of Schaffer collaterals and commissural fibers located in the stratum radiatum. The NMDARs-mediated component was isolated from slices perfused with low magnesium aCSF (0.1 mM to relieve NMDARs of their Mg^2+^ blockade) and supplemented with NBQX (10 μM). The slope of three successive PFVs and fEPSPs was averaged using WinLTP^®^ software 2.30 [50,51]. To evaluate the level of NMDAR activation, an index of synaptic efficacy (Ise) corresponding to the fEPSP/PFV ratio was plotted against increasing stimulus intensities (400 and 500 µA). The effects of successive application of SB-271046 with increased concentrations (0.1 µM, 1 µM, and 10 µM) in the control or bicuculline-supplemented aCSF (10 µM) were evaluated by comparing Ise before and 15 min after the addition of the drug to the medium.

After the evaluation of the effective dose of SB-271046 on NMDARs’ activation, the effects on basal synaptic transmission and synaptic plasticity were assessed using extracellular recordings obtained with a Multi-Electrodes array (MEA2100, Multi-Channel System, 60MEA200/30iR-Ti 8*8, Reutlingen, Germany). The recording chamber was equipped with 60 recording electrodes (30 µm diameter with 200 µm spacing). To assess basal neurotransmission, fEPSPs also evoked in the CA1 area by stimulation of the Schaffer collaterals/commissural fibers were measured using LTP director software^®^ 1.4 (Multi-Channel Systems MCS). Input-output (I/O) curves were obtained with voltage currents ranging from 3000 to 5000 mV. The paired-pulse facilitation (PPF) of basal synaptic transmission was investigated through paired electrical stimulation with an interstimulus interval of 30 ms. The voltage current selected to induce PPF corresponded to a response of 50% of the maximal achievable fEPSP determined with the I/O curve. The PPF was calculated as the ratio of the slope of the second stimulation to the first one. In both the I/O and PPF experiments, the effects of SB-271046 were evaluated by comparing the values before and 15 min after the addition of the antagonist to the aCSF. To investigate the short-term potentiation and long-term potentiation (STP and LTP, respectively) of synaptic transmission, a test stimulus was applied every 20 s and adjusted to obtain an fEPSP with a baseline slope of 0.1 V/s. After 15 min of stable baseline, a conditioning stimulus was applied consisting of either a high-frequency stimulation (HFS, 1 burst of stimulations delivered at 100 Hz for 1 s) or a theta-burst stimulation (TBS, 10 bursts of 4 pulses delivered at 100 Hz separated by 200 ms and repeated twice with an interval of 20 s). In both HFS and TBS experiments, testing with a single pulse was then resumed for 60 min. For statistical analyses, STP was considered as changes occurring between 5 and 15 min after the conditioning stimulation while LTP was considered as occurring between 45 and 60 min.

### 2.4. Data Analysis

Analyses were performed with R^®^ software 4.2.0 and GraphPad^®^ Prism 8.3. The assessment of normality of the data distribution was performed using the Shapiro test and the homogeneity of variances with Levene’s test. When data were normally distributed, two-way analysis of variance (ANOVAs with repeated measures or not) were used for group comparisons, and data are expressed as mean + standard error of the mean (SEM). Otherwise, data are expressed in median ± interquartile. To assess drugs effects, paired *t*-tests (for data normally distributed) or Wilcoxon paired tests (for data not normally distributed) were used. A significant difference was admitted in all cases when *p* was below 0.05.

## 3. Results

### 3.1. Effects of SB-271046 on NMDARs’ Activation

NMDARs’ activation is critical for functional plasticity in the neuronal networks that drive memory formation [37,38,39,40,41]. To accurately determine possible dose-dependent effects of 5-HT6Rs’ antagonism, we looked first at changes in pharmacologically isolated NMDARs-mediated fEPSPS (see Section 2) that were induced by increased concentrations of SB-271046.

In almost all experimental subgroups (Figure 1), the NMDARs’ activation determined by the Ise index (corresponding to the fEPSP/PFV ratio) was significantly increased by SB-271046. This significant increase was found in male (Figure 1A) and female mice (Figure 1B) and was not dose-dependent.

Because 5-HT6Rs have been suggested to modulate GABA inhibition [32,52,53], we next monitored the effects of SB-271046 on NMDARs’ activation in the presence of the GABAARs’ antagonist bicuculline (Figure 2). In these additive experiments, the 5-HT6Rs’ antagonist was applied at 10 µM to induce the reproducible increase in NMDARs activation found in the control conditions. On the one hand, we did not observe any detectable effect of bicuculline (10 µM) prior to SB-271046 application in male mice; this held true regardless of the intensity of stimulation. Conversely, a significant increase was found in female mice but only for the 500 µA stimulation intensity. On the other hand, the increase in NMDARs’ activation by SB-271046 was suppressed by bicuculline in male mice (Figure 2A), whereas a significant enhancement persisted in female mice (Figure 2B).

### 3.2. Effects of SB-271046 on Basal Synaptic Transmission

We next determined whether the 5-HT6Rs antagonist could also alter the non-NMDARs-mediated fEPSPs supporting basal synaptic transmission. In these experiments, we assessed the effects of 10 µm SB-271046 on the fEPSPs induced in the control aCSF by intensities of stimulation ranging from 3000 to 5000 mV. A significant increase in non-NMDARs fEPSPs by SB-271046 was found compared with that produced by the vehicle treatment in both male and female mice regardless of the intensity of the stimulation (Figure 3A).

Basal neurotransmission depends on the presynaptic release of glutamate, which can be experimentally assessed by the paired-pulse facilitation (PPF) paradigm (see [54,55]). In both male and female mice, a significant PPF was induced using paired stimulation separated by a 30 ms interval, which was not statistically different between the vehicle and SB- 271046 (Figure 3B).

### 3.3. Effects of SB-271046 on Functional Plasticity

Considering the potentiation effect of SB-271046 on NMDARs’ activation, we finally assessed NMDARs-dependent functional plasticity using high-frequency stimulation (HFS) and theta-burst stimulation (TBS) in hippocampal slices from male and female mice. In both HFS- and TBS-related experiments (Figure 4), the mean fEPSPs’ magnitude (percent of increase relative to the baseline, from 5 to 15 min and 45 to 60 min after the tetanus for short-term potentiation (STP) and long-term potentiation (LTP), respectively) was significantly increased indicating that STP and LTP were promoted in all cases. For both HFS- and TBS-related protocols (Figure 4A,B), the magnitudes of STP and LTP did not differ between the vehicle and SB-271046 (10 µM) treatments (Table 1).

## 4. Discussion

This electrophysiological study aimed to investigate the effects of the blockade of subtype 6 of the serotoninergic receptors (5-HT6Rs) on the synaptic activity of CA1 neuronal networks in ex vivo hippocampal slice preparations of male and female mice. It was found that in both sexes, the 5-HT6R antagonist SB-271046 increases synaptic-glutamate-related synaptic potentials mediated by NMDARs and non-NMDARs without affecting glutamate release. Interestingly, the NMDARs-related improvement was prevented by GABAARs blockade only in male mice, indicating a sex-dependent involvement of inhibitory transmission. Finally, this study found that although 5-HT6Rs blockade increases NMDARs activation, it does not significantly impact the related NMDARs-dependent functional plasticity at CA3/CA1 synapses. 

Cloned in 1993 [7], the 5-HT6Rs, which belong to the family of Gs-protein-coupled receptors, are located in brain regions involved in cognitive functions such as the cortex, striatum, or hippocampus. This localization allows the identification of these receptors as putative new targets for the development of alternative treatments aimed at preventing cognitive-related disabilities [1,3,7,36,56]. The pharmacological blockade of 5-HT6Rs by the selective antagonist SB-271046 [57] leads to increases in the dopamine, acetylcholine, and glutamate levels in the hippocampus of rats [28,29,30,31] and improves memory performances in numerous behavioral paradigms [15,19,20,22,58]. The promnesiant effect of 5-HT6Rs’ blockade on recognition memory has been shown to be dependent on close interactions with the glutamatergic system in mice [27] and rats [59], though the molecular mechanisms still remain unclear. Our electrophysiological study provides functional evidence for a tonic modulation of glutamate-related synaptic activity by 5-HT6Rs in hippocampal neuronal networks. Indeed, we report that the low-frequency-induced synaptic potentials mediated by non-NMDARs as well as NMDARs are increased by antagonism of 5-HT6Rs. These results are in line with previous data of whole-cell patch-clamp electrophysiological recordings performed in the medium spiny neurons of the rat striatum. Indeed, this investigation revealed that, conversely, the activation of 5-HT6Rs by the agonist ST1936 reduced the amplitude of spontaneous glutamatergic postsynaptic currents [60]. Additionally, in agreement with this study, we found that the pharmacological manipulation of 5-HT6Rs does not alter the paired-pulse facilitation of synaptic transmission, a form of short-term plasticity used for assessing the calcium-dependent mechanisms of transmitter release. This result therefore indicates that 5-HT6Rs do not modulate glutamate transmission by changing synaptic glutamate release but rather by activating postsynaptic mechanisms in the neuronal network. Accordingly, we found that the facilitation effect of 5-HT6Rs’ antagonism on NMDARs activation is prevented by GABAARs blockade, suggesting changes in the excitation/inhibition balance. However, further experiments are still needed to detail the underlying mechanisms, including patch-clamp recordings, as performed by West et al. (2009), whose results are discussed below [42].

Several studies have investigated 5-HT6Rs’ localization on cell types, reporting that, in the hippocampus, these receptors are mainly expressed on excitatory pyramidal cells and on inhibitory GABA interneurons, though to a much weaker degree [3,33,35]. According to this presence on inhibitory cells, the agonist WAY-181187 increases the frequency of spontaneous inhibitory postsynaptic currents in the CA1 hippocampal area, an effect that is prevented by the selective 5-HT6Rs antagonist SB-399885 [42]. Interestingly, we provide here, for the first-time, evidence for sexual dimorphism, given that the 5-HT6Rs’ antagonist affects NMDARs’ activation through GABA modulation in male but not in female mice. Unfortunately, the morphological studies available so far have been conducted in male rodents and are therefore not informative for a possible sex difference in the cellular localization of 5-HT6Rs. Regardless, we can hypothesize that 5-HT6Rs rather indirectly control glutamate transmission in males through the modulation of the inhibitory tone, whereas they can directly act on pyramidal cells in females through changes in membrane excitability. In our opinion, this is an important observation because knowledge on sexual dimorphism in the healthy brain is of growing interest [43,61,62] and because sex dichotomy is often associated with brain disorders [63,64,65,66], particularly in the case of schizophrenia [45,46]. Interestingly, a clinical investigation using the 5-HT6Rs’ agonist inhibitor avisetron (AVN-211) reported the benefits of the drug only in women schizophrenia patients with residual psychotic symptoms [49]. The conclusion of this clinical trial, emphasizing the impact of the patients’ sex on the treatment response to the serotonergic compound, perfectly fits with the results of the present study, which reveal distinct 5-HT6Rs-related mechanisms driven in male and female mice. In addition, different levels of response to antipsychotic drugs (for example, clozapine, olanzapine and risperidone), used more conventionally to treat patients, were also found between men and women. Differences were observed for the effective dosage of antipsychotic drug to obtain a response, being usually higher in men than women, and the side effects of antipsychotics were more severe for women (for a review, see [47,48,67]). Thus, sex differences should be considered with caution in future clinical and preclinical studies, not only because there are differences in treatment responses but also because variability in disease onset, disease progression, and symptom onset are usually observed between men and women with schizophrenia [44]. Furthermore, several preclinical models of schizophrenia found sex-specific alterations in cognitive functions [45,68,69]. Our results provide evidence that this dichotomy is already present at the neuronal level, which should be taken into consideration for the development of a selective treatment. 

Finally, our study shows that though the 5-HT6Rs antagonist increases the low -frequency-induced NMDARs synaptic potentials, it has no significant effect on the NMDARs-dependent LTP generated at the same synapses by high-frequency stimulation (either by HFS but also TBS which is more sensitive to variations in GABAergic tone than HFS). Our result could be contradictory with the attenuation of TBS-induced LTP promoted by the 5-HT6R agonist WAY-181187 at the same neuronal connections [42]. However, this latter study also indicated that the 5-HT6R antagonist SB-399885 tested alone did not affect LTP expression, as we found here with SB-271046. In addition to being used for preventing the effects of an agonist, the interest in testing an antagonist is to investigate whether the tonic activity of a specific receptor subtype is involved in the control of neuronal network activity or plasticity. From the results of the present study, we can therefore argue for a low, if any, tonic influence of 5-HT6Rs on functional synaptic plasticity, at least in the hippocampal slice preparations. 

To conclude, the present study provides new informative data related to the synaptic interactions between 5-HT6Rs and glutamate neurotransmission in the hippocampus. We highlight the differential modulation by 5-HT6Rs of the excitatory/inhibitory balance within neuronal networks where male or female subjects are concerned. Indeed, we found an increase in the hippocampal network activity in male mice in particular through an action on the inhibitory tone, whereas a greater involvement of glutamatergic neurotransmission could be selective to female mice. These data could help in initiating new pharmacological strategies to treat the sex-dependent cognitive impairments notably associated with schizophrenia.

## Figures and Tables

**Figure 1 biomolecules-13-00751-f001:**
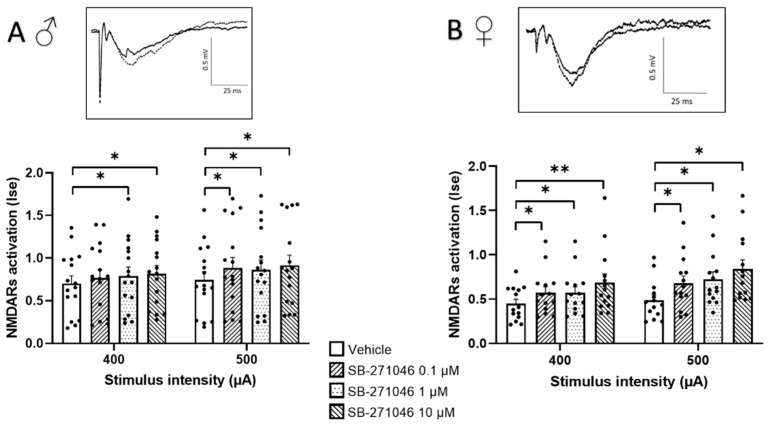
SB-271046 increases NMDARs activation in CA1 hippocampal networks. Effects of increasing concentrations of SB-271046 (0.1 µM, 1 µM, and 10 µM) on Ise of isolated NMDARs-mediated fEPSPs determined at 400 µA and 500 µA stimulation intensities in male (**A**) (mean + SEM, *n* = 16 slices, * *p* < 0.05, paired *t*-tests) and female (**B**) mice (mean + SEM, *n* = 14 slices, * *p* < 0.05 and ** < 0.001, paired *t*-tests). Treatment effect was significant for male and female (* *p* < 0.05 two-way ANOVA repeated measures) mice. The synaptic efficacy (Ise) corresponds to the fEPSP/PFV ratio. In inserts are representative examples of superimposed traces recorded for 400 µA stimulation intensity in vehicle (filled line) and SB-271046 (dashed line).

**Figure 2 biomolecules-13-00751-f002:**
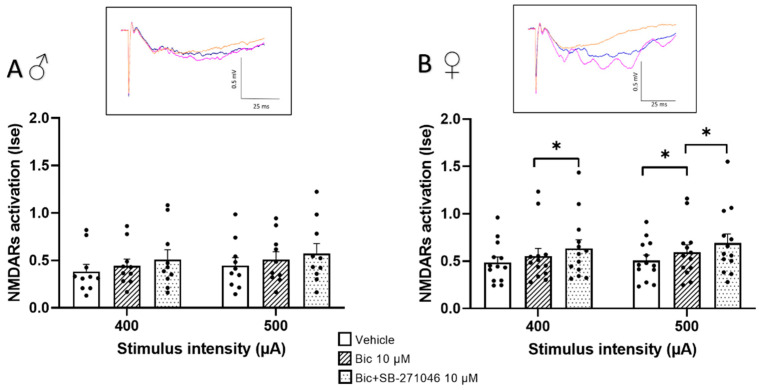
The potentiating effect of SB-271046 on NMDARs’ activation involves GABAergic transmission in male but not female mice. Effects of SB-271046 application (10 µM) on synaptic efficacy (Ise) of NMDARs-mediated fEPSPs determined at 400 and 500 µA stimulation intensities in the presence of the GABAARs antagonist bicuculline (Bic, 10 µM) in male (**A**) (mean + SEM, *n* = 10 slices) and female (**B**) mice (mean + SEM, *n* = 13 slice, * *p* < 0.05, paired *t*-tests). In inserts are representative examples of superimposed traces recorded in vehicle aCSF (orange line), in the presence of bicuculline before (blue line) and after addition of SB-271046 (pink line).

**Figure 3 biomolecules-13-00751-f003:**
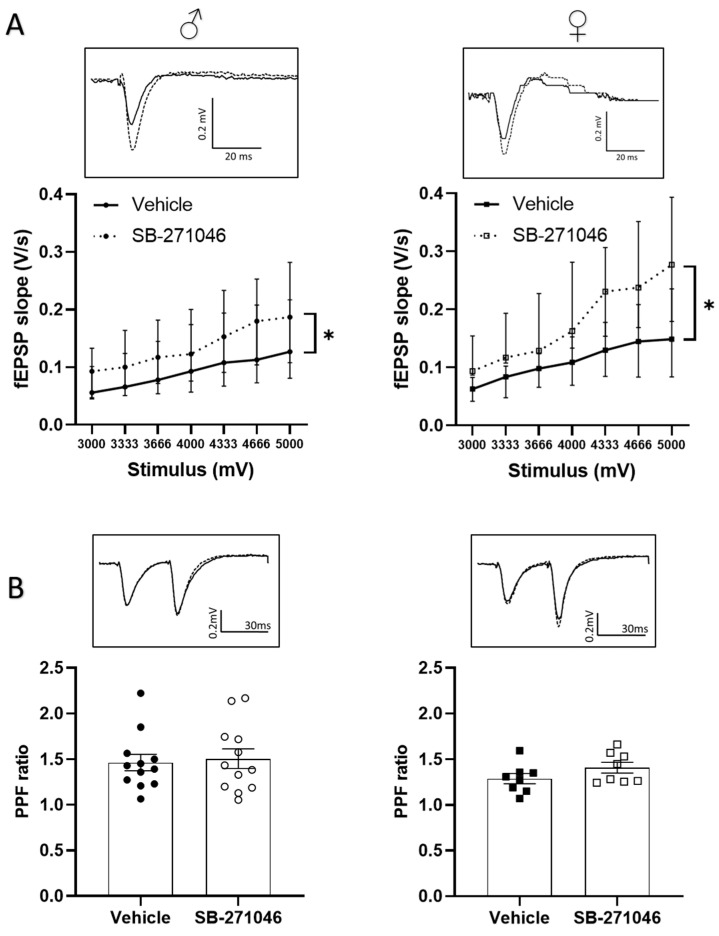
SB-271046 increases basal transmission without affecting mechanisms underlying presynaptic release of glutamate (**A**) Effect of SB-271046 (10 µM, dotted line) vs. vehicle (full line) on non-NMDARs-mediated fEPSPs (median ± interquartile range) induced by increasing stimulus voltages in male (left, *n* = 11 slices, * *p* < 0.05, Wilcoxon paired test) and female mice (right, *n* = 8 slices, Wilcoxon paired test: * *p* < 0.05). (**B**) Paired pulse facilitation (PPF, mean ± SEM) ratio calculated in presence of SB-271046 (hollow circle and square dots) or vehicle (black circle and square dots) in male (left, *n* = 12 slices) and female mice (right, *n* = 8 slices). In inserts are representative examples of superimposed traces recorded for 4000 mV stimulus voltages in vehicle (filled line) and SB-271046 (dashed line).

**Figure 4 biomolecules-13-00751-f004:**
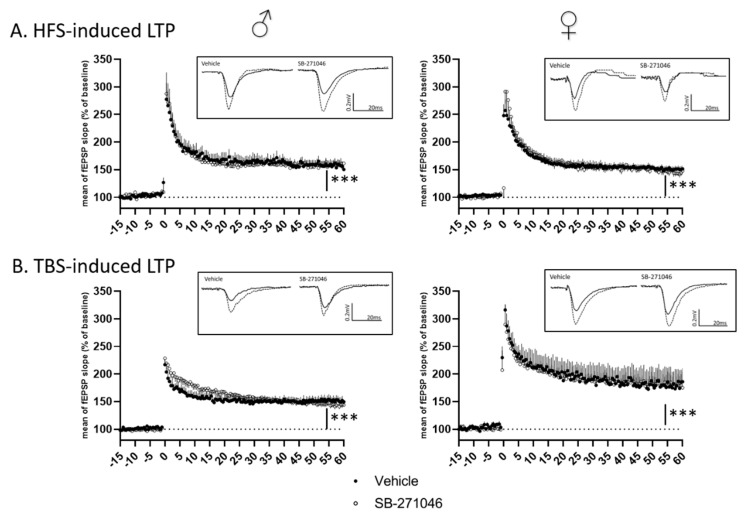
Long-term potentiation is not altered by SB-271046. (**A**) Time-course of high-frequency stimulation (HFS)-induced long-term potentiation (LTP) in vehicle and in presence of SB-271046 (10 µM) in male (left, vehicle *n* = 9 slices vs. SB-271046 *n* = 8 slices) and female (right, vehicle *n* = 12 slices vs. SB-271046 *n* = 10 slices) mice. (**B**) Time course of theta-burst stimulation (TBS)-induced LTP in vehicle and in presence of SB-271046 (10 µM) in male (left, vehicle *n* = 10 slices vs. SB-271046 *n* = 9 slices) and female (right, vehicle *n* = 5 slices vs. SB-271046 *n* = 5 slices) mice. Vehicle condition is represented by black dots and SB-271046 by white dots. Short-term potentiation (5–15 min; STP) and LTP (45–60 min) are promoted by both HFS and TBS in male and female mice compared with baseline (*** *p* < 0.0001, two-way ANOVA, time effect). In inserts are representative examples of superimposed traces recorded before (filled line) and 60 min (dashed line) after the conditioning stimulation in vehicle or SB-271046 conditions.

**Table 1 biomolecules-13-00751-t001:** SB-271046 has no effect on the magnitude of short- or long-term potentiation. Mean values of the magnitude of high-frequency stimulation (HFS)- and theta-burst stimulation (TBS)-induced short-term potentiation (5–15 min; STP) and long-term potentiation (45–60 min; LTP) in control and SB-271046-supplemented aCSF (10 µM) in male and female mice.

		HFS		TBS	
		STP	LTP	STP	LTP
Male	Vehicle	180.6 ± 13.35%	158.2 ± 8.45%	160.5 ± 8.97%	180.1 ± 17.67%
	SB-271046	174.5 ± 8.56%	157.6 ± 8.13%	177.0 ± 6.91%	181.2 ± 28.48%
Female	Vehicle	166.6 ± 6.50%	148.0 ± 2.87%	216.8 ± 5.62%	151.1 ± 7.62%
	SB-271046	181.7.5 ± 8.85%	148.5 ± 4.44%	211.7 ± 28.76%	146.2 ± 4.48%

## Data Availability

The data presented in this study are openly available at [https://doi.org/10.5281/zenodo.7864762].

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
