# Peer review of "5-HT6 Receptors Sex-Dependently Modulate Hippocampal Synaptic Activity through GABA Inhibition"

_biomolecules, 2023, doi:10.3390/biom13050751_

Round 1

Reviewer 1 Report

 Remarks:

Abstract is clearly written. Introduction is fine.

Methods:

line 93: fEPSPs – does not explained (field excitatory postsynaptic potentials (fEPSPs),

as well as == γ-Aminobutyric acid type A receptors (GABAARs)

– line 78.

line 105 -  using …. using (?) were assessed using extracellular recordings obtained using Multi-Electrodes array

Statistical analysis is relevant.

 Results:

line 152 – do not need “either” (remove it) - This increase was found either in male and female mice

 Figure 1 and 2 are fine.

 line 195: between vehicle SB-and 271046 (??) I suppose it should be: …between vehicle and SB- 271046

please correct!

line 200-201 – please close the bracket after?? (median ± interquartile range

line 209-210: probably here you should not repeat the meanings of HFS and TBS? as it was done on the lines 121-122 in Methods section.

Discussion:

 line 248: glutamate levels in the hippocampus in rat - glutamate levels in the hippocampus of rats

line 301: the interest for testing an antagonist -  change on :…the interest for testing of an antagonist

  The last paragraph is formatted with different space compare o previous (line 305)

References is OK.

Reviewer 2 Report

Dear editor,

I have carefully reviewed the manuscript which I consider suitable for publication in this form. However, I would like to suggest to the authors to report or better introduce a figure with the chemical structures of the compounds whose pharmacological activity is described. I think it could be interesting in order to establish a correlation between structure and specific activity.

Best Regards

Reviewer 3 Report

In the current study authors examined the role of 5-HT6 receptors and by using antagonist of 5-HT6 receptor, they found that both basal excitatory synaptic transmission and isolated N-methyl-D-aspartate receptors (NMDARs) activation on the Ca1-CA3 synapses were significantly increased by the inhibitor. It is an interesting study but there are several issues are needed to be considered:

1.In the conclusion authors mentioned about sex-dependent 5-HT6Rs effect on synaptic activity at CA3/CA1 hippocampal connections. However, from title of manuscript it is missing.

2.Authors mentioned 5-HT6 receptor changes has the role in the excitation/inhibition balance. However, they only analyzed excitatory NMDA synapses. By just using GABAAR antagonist, we cannot conclude for excitatory/inhibitory imbalance, it is important to measure synaptic activity on both excitatory and inhibitory synapses.

3. The focus of this study was on 5-HT6 receptor due to its critical role in memory function, however authors did not run any behavioral analysis to correlate their findings with the memory and conclude about sex-dependent role of 5-HT6 receptor in memory function. Without behavioral outcome, findings of current study will not be novel.

Round 2

Reviewer 3 Report

Authors responded to the comments in a satisfactory manner.